# Experience of Isavuconazole as a Salvage Therapy in Chronic Pulmonary Fungal Disease

**DOI:** 10.3390/jof8040362

**Published:** 2022-03-31

**Authors:** Lisa Nwankwo, Desmond Gilmartin, Sheila Matharu, Ali Nuh, Jackie Donovan, Darius Armstrong-James, Anand Shah

**Affiliations:** 1Pharmacy Department, Royal Brompton Hospital, Guy’s and St. Thomas’ NHS Foundation Trust, London SW3 6NP, UK; 2Clinical Informatics, Royal Brompton and Harefield Hospital Foundation NHS Trust, Fulham, London SW3 6HP, UK; dgilmartin@gmx.com (D.G.); sheila.matharu@rmh.nhs.uk (S.M.); 3Microbiology Department, Royal Brompton Hospital, Guy’s and St. Thomas’ NHS Foundation Trust, London SW3 6NP, UK; a.nuh@rbht.nhs.uk (A.N.); d.armstrong-james@rbht.nhs.uk (D.A.-J.); 4Department of Pathology, Royal Brompton and Harefield NHS Foundation Trust, London SW3 6NP, UK; j.donovan@rbht.nhs.uk; 5MRC Centre for Molecular Bacteriology and Infection, Department of Infectious Diseases, Imperial College London, London SW7 2AZ, UK; 6Department of Respiratory Medicine, Royal Brompton Hospital, Guy’s and St. Thomas’ NHS Foundation Trust, London SW3 6NP, UK; 7MRC Centre of Global Infectious Disease Analysis, Department of Infectious Disease Epidemiology, School of Public Health, Imperial College, London W2 1PG, UK

**Keywords:** antifungal resistance, isavuconazole, cystic fibrosis, pulmonary disease, aspergillus fumigatus, pulmonary aspergillosis, respiratory disease, antifungal stewardship, therapeutic drug monitoring, minimum inhibitory concentration, MIC

## Abstract

**Background:** Instances of resistant fungal infection are rising in pulmonary disease, with limited therapeutic options. Therapeutic drug monitoring of azole antifungals has been necessary to ensure safety and efficacy but is considered unnecessary for the newest triazole isavuconazole. **Aims:** To characterise the prevalence of isavuconazole resistance and use in a tertiary respiratory centre. **Methods:** A retrospective observational analysis (2016–2021) of adult respiratory patients analysing fungal culture, therapeutic drug monitoring, and outcome post-isavuconazole therapy. **Results**: During the study period, isavuconazole susceptibility testing was performed on 26 *Aspergillus* spp. isolates. A total of 80.8% of *A. fumigatus* isolates had isavuconazole (MIC > 1 mg/L, and 73.0% > 2 mg/L) with a good correlation to voriconazole MIC (r = 0.7, *p* = 0.0002). A total of 54 patients underwent isavuconazole therapy during the study period (median duration 234 days (IQR: 24–499)). A total of 67% of patients tolerated isavuconazole, despite prior azole toxicity in 61.8%, with increased age *(r_pb_* = 0.31; *p* = 0.021) and male sex (φ_c_ = 0.30; *p* = 0.027) being associated with toxicity. A total of 132 isavuconazole levels were performed with 94.8% > 1 mg/L and 72% > 2 mg/L. Dose change from manufacturer’s recommendation was, however, required in 9.3% to achieve a concentration of >2 mg/L. **Conclusion**: We describe the use of isavuconazole as a salvage therapy in a chronic pulmonary fungal disease setting with a high prevalence of azole resistance. Therapeutic concentrations at standard dosing were high; however, results reinforce antifungal stewardship for optimization.

## 1. Introduction

The burden of fungal infection in patients with underlying chronic lung disease is increasing [1]. This is primarily driven by the environmental mould *Aspergillus fumigatus*, and other filamentous fungi [2]. There is a spectrum of clinical presentation of pulmonary aspergillosis depending on the host immune response and/or the presence of pre-existing lung disease. This ranges from sensitisation and allergic bronchopulmonary aspergillosis (ABPA), to chronic pulmonary aspergillosis (CPA) and invasive fungal disease [3]. The global estimate of patients living with chronic pulmonary fungal disease is substantial, with ~5 million individuals with ABPA and ~3 million individuals with CPA, respectively [4,5].

Despite the significant burden of chronic respiratory fungal disease, there are only five classes of antifungal agents (polyenes, triazoles, echinocandins, pyrimidines, and allyamines). The triazoles (itraconazole, voriconazole, posaconazole, and isavuconazole) are the only mould-active antifungal drugs available in oral formulation. Prolonged usage is often associated with toxicity and complicated by significant drug–drug interactions, and response rates are highly variable. A recent, prospective cohort study in chronic pulmonary aspergillosis showed ~30% of patients had to stop or change triazole therapy due to toxicity [6]. 

Therapeutic drug monitoring (TDM) has been necessary to ensure the safety and efficacy of itraconazole, voriconazole, and posaconazole [7,8,9] but is considered unnecessary for the newest triazole, isavuconazole [9], the use of which is increasing. TDM is typically necessary where standard dosing gives unpredictable responses, or where there are established dose–exposure–to–response relationships, resulting in treatment failure due to either efficacy or toxicity [7,10]. In patients with invasive fungal infections (IFIs), subtherapeutic serum–drug concentrations are associated with breakthrough infection or disease progression of IFIs [11]. Within chronic pulmonary fungal disease, we have previously shown a relationship between drug exposure and treatment outcome for ABPA in patients with cystic fibrosis [12]. The additional, high-prevalence and emergence of antifungal resistance rates seen in chronic pulmonary fungal disease [13,14,15,16,17] alongside high treatment failure rates suggest optimising therapeutic drug exposure may be critical to achieving optimal outcomes [7,8]. 

Itraconazole and voriconazole have been noted to have high inter- and intra-patient variability with little or no correlation between dose and plasma level [18,19,20,21,22] Posaconazole has improved pharmacodynamic/pharmacokinetic properties with the gastro-resistant-modified-release formulation [23]; however, antifungal resistance often mirrors that of itraconazole [13], which is prevalent in a chronic pulmonary fungal setting [24]. Drug–drug interactions, however, remain a significant issue with these triazoles as CYP3A4 inhibitors [25,26,27]. This is particularly relevant in patients with co-existent mycobacterial infection or CF with the advent of novel CFTR-modulator therapies that are metabolized by the cytochrome P450 (CYP) pathway [28]. 

Isavuconazole is the newest addition to this class of triazole antifungals. Isavuconazole is structurally similar to voriconazole, whilst posaconazole is structurally similar to itraconazole [29]. Isavuconazole received approval in 2015 from the European Medicines Agency (EMA) in Europe and the Food and Drug Administration (FDA) in the USA [30,31,32] to treat invasive aspergillosis and invasive mucormycosis in adult patients. It has the advantages of once-a-day dosing, a good pharmacokinetic and pharmacodynamic profile, and a lower propensity for drug interactions; however, some intra- and inter-patient variability has been observed [33,34]. Currently, it is used in tertiary respiratory practice predominantly as a salvage therapy due to toxicity or treatment failure related to alternative azole use [35]. Previous clinical trial data in an invasive fungal disease setting did not demonstrate a relationship between isavuconazole drug exposure and efficacy (clinical response, all-cause mortality) or safety endpoints [36]. To date, there have been no studies of isavuconazole TDM performed in a chronic pulmonary fungal setting where there is increased prevalence of antifungal resistance, drug–drug interactions, and cohorts with known reduced bioavailability (e.g., CF) [9]. 

In this retrospective observational cohort study, we analyse the isavuconazole susceptibility of *Aspergillus fumigatus* isolates in a tertiary respiratory referral centre to understand the prevalence of isavuconazole antimicrobial resistance. In addition, we analysed isavuconazole tolerability and therapeutic drug levels. Based on our findings, we propose guidance on therapeutic drug monitoring for isavuconazole use in a chronic respiratory fungal disease setting.

## 2. Materials and Methods

This was a retrospective observational cohort analysis of adult patients at a specialist tertiary respiratory centre over a 5-year period (September 2016–August 2021). Retrospective electronic health record data collection and protocols were approved by the UK Research Ethics Committee (REC reference:18/HRA/1074). Patients who had either received isavuconazole treatment and/or had isavuconazole drug susceptibility testing on filamentous pathogenic mould isolates were included. Data was obtained from trust electronic prescribing management administration systems following data integration and consequential data mining using SAS Enterprise Guide software. Fungal culture results, isolate antifungal sensitivities, and azole minimum inhibitory concentration (MIC) were recorded, in addition to trough isavuconazole serum levels. Clinical demographics, including age, sex, disease-specific information, and co-morbidities, were collected. Case records were reviewed to analyse drug toxicities (attributed to the drug by the clinical team) and reasons for discontinuation or change of azole therapy. 

Azole-resistant isolates were confirmed with a standard microbroth dilution method according to Clinical and Laboratory Standards Institute (CLSI) reference guidelines [37]. In our centre, antifungal susceptibility testing for isavuconazole was carried out where requested by the microbiologist/mycologist when resistance to first-line therapies (itraconazole/voriconazole) was encountered or where isavuconazole therapy was being considered. Sensitivity testing for the antifungals was determined using the two commercially available antifungal susceptibility testing kits, Sensititre™ YeastOne™ (Thermo Fisher) and MICRONAUT-AM (MERLIN Diagnostika) [38]. Isavuconazole is absent from these commercially available antifungal susceptibility testing kits, and thus isavuconazole MIC was determined by the UK Mycology Reference Laboratory in accordance with the CLSI broth microdilution method [39]. Isolates with an epidemiological cut-off (ECOFF) MIC value > 1 mg/L were considered non-wild type (NWT) (CLSI) [40]. Isavuconazole trough drug levels on serum were carried out as part of the standard of care at our institution. Isavuconazole testing is not commonly performed across the country, and only a few centres and the reference laboratories perform this test. Isavuconazole was measured using 2-DTurboFlow™ high-performance liquid chromatography, tandem mass spectrometry (2D HPLC–MS/MS) (Thermo Fisher Scientific™ Prelude SPLC™ system and Thermo Fisher Scientific™ SQ Endura™). Analysis was carried out by hospitals’ laboratories using in-house methodology, developed in-line with FDA and EMA guidelines. The intra- and inter-assay precision across the reporting range of 0.2 mg/L to 8 mg/L was below 5.2%. Samples were stored at −20 °C prior to analysis. Extraction was via protein precipitation in acetonitrile with isuvaconazole-d4 as the internal standard. Chromatographic separation was achieved using gradient chromatography with mobile phases 10 nM ammonium acetate UP-H_2_O and 10 nM ammonium acetate methanol, and HPLC equipment were used thereafter. Mass spectrometric analysis was conducted in positive ion mode using atmospheric chemical pressure ionisation with mass transitions 438.080 to 224,369.071. Isavuconazole has a long half-life of 130 h [36], and the trough sample is taken at steady state i.e., 3–4 weeks, sampled just before the patient takes their morning dose. 

Statistical analysis was performed using GraphPad Prism 9.3.0 software and Python version 3.8. Tests for normality were performed using Shapiro–Wilk and D’Agostino–Pearson tests. Pearson’s correlation tests were used for MIC correlation analysis (continuous, parametric variables). The outcome variable “Adverse Drug Reaction–Y/N” was coded as 1/0, and the point biserial correlation coefficient (*r_pb_*) was used to determine the correlation between this dichotomous variable and the continuous variables of age, average, TDM value, and duration of therapy in days. Pearson’s chi-square test (χ^2^) of independence was used to test whether there was an association with the two categorical variables of sex (male/female) and adverse drug reaction (Y/N). Where the *p* value was <0.05, Cramér’s phi φ_c_ was used to determine the strength of the association. Results were presented as mean ± standard deviation for parametric variables and median (interquartile range) for non-parametric variables. *p*-value < 0.05 was considered significant. 

## 3. Results

Within the study period, 850 mould cultures were isolated from respiratory samples in 363 patients. *Aspergillus fumigatus* was the most predominant isolate at 79.65%. Of these, isavuconazole sensitivity testing was performed on 29 samples from 21 patients (Appendix A).

### 3.1. Isavuconazole Susceptibility Testing

Within the study period, isavuconazole susceptibility testing was performed on 26 *Aspergillus* spp. isolates. Figure 1 shows the distribution of MIC for *Aspergillus fumigatus* in the total cohort and also in individuals where isavuconazole therapy was used.

Figure 1A–D shows the distribution of triazole MIC with the CLSI and EUCAST ECOFF values in all those cases that received MIC testing. A total of 73% of isolates had a MIC above the EUCAST ECOFF or ATU, and 80.8% of isolates were above the CLSI ECOFF, i.e., non-wild-type for isavuconazole. There was good correlation between isavuconazole and voriconazole MICs (Pearson’s r = 0.7, *p* = 0.0002 (95% CI 0.43–0.85)), (Figure 2). MIC levels for non-*Aspergillus* moulds, including *Lomentospora, Rasamsonia,* and *Exophiala* organisms are shown in Appendix A.

### 3.2. TDM and Tolerability of Isavuconazole during Study Period

A total of 54 patients received isavuconazole therapy over the course of the study period, with patient demographics shown in Table 1. Intolerance to other azoles was the predominant reason for isavuconazole use (34 patients, 61.8%), with prior alternate azole use being high (97%), given its use as a salvage therapy. Isavuconazole was used in 11 patients (20%) due to treatment failure defined by radiological or serological progression despite alternate azole use (Table 1).

Dose variations from standard manufacturer’s dose recommendations were seen in 13 (24%) patients. In five (9.3%) cases, dose escalations in response to lower TDM values or high MIC levels of pathogenic isolates were made, with dose reductions in eight patients (14.8%) due to intolerance or supratherapeutic levels. The maximum tolerated dose used in this study was 400 mg daily, used in one patient until the patient stopped due to pregnancy. Lower, cautious initiation dosing (100 mg daily after loading) was used in 11 patients where significant toxicity was seen on previous triazole therapy, or due to higher frailty. Of these, five were still intolerant of isavuconazole, despite the lower-than-standard initiation dose (Table 2). Figure 3 shows outcome whilst on isavuconazole treatment during the study period. In 2 (5%) individuals, therapy was stopped due to disease resolution/stability, and 28 (52%) individuals continued with evidence of disease stability. Median treatment duration was 234 days (IQR 24–499).

Despite a cohort with very high prior intolerance to azole therapy, isavuconazole treatment was tolerated in 66.6% (*n* = 36) of patients. Attributable drug toxicity was nevertheless still observed in 33.3% (18 of 54 patients) (Table 2), with hepatotoxicity, skin reactions, and fatigue/drowsiness being the predominant side effect(s), all of which were serious enough to necessitate discontinuation of the therapy. 

Of note, significant hair loss was not observed despite having been a prominent feature in some patients previously on triazole therapy. Increased age *(r_pb_* = 0.31; *p* = 0.021) and male sex (Cramér’s phi coefficient φ_c_ = 0.30; *p* = 0.027) were associated with increased risk for the development of adverse effects (Table 3). Therapy was discontinued in only 7.4% (four patients) due to disease progression. Six patients died during the study period, of which two were attributed directly to their pre-existing underlying fungal infection with respiratory failure as the cause of death. The other patients died due to complications from their underlying lung disease. 

In individuals who received isavuconazole therapy, *Aspergillus fumigatu**s* was the most frequent fungal isolate (*n* = 39, 73.5%) (Table 1). Other isolates grown included 1 *Apergillus niger*, 5 *Exophiala dermiditidis*, 1 *Lomentospora prolificans*, 2 *Rasamsonia argillacea*, 1 *Aspergillus terreus*, and 6 *Scedosporium apiospermum*. Where isavuconazole was used as a salvage therapy in individuals with chronic lung disease (all with prolonged previous azole exposure), 73.0% of *Aspergillus fumigatus* isolates had an isavuconazole MIC above EUCAST ECOFF of >2 mg/L, and 80.8% of isolates were >1 mg/L, the CLSI ECOFF (Figure 1E). 

Due to the manufacturer not recommending routine TDM testing, and unfamiliarity among clinicians with use, not all patients underwent TDM, and some patients received >1 blood concentration evaluation of trough levels, usually in response to raised liver function tests, or super- or supratherapeutic levels. In patients receiving >6 months of isavuconazole, routine monitoring accounted for the >1 TDM performed in a subject. Within the study period, 44 (81%) patients treated with isavuconazole had therapeutic trough drug levels performed, with a total of 132 drug levels measured. Of these, 86% (36 patients) attained TDM levels greater than 2 mg/L. Figure 4 shows the distribution of isavuconazole trough levels attained during the study period with a median isavuconazole level of 2.77 mg/L (IQR 2.35–3.10). Intra-individual variation was 2.03 mg/L (IQR 0.56–2.83). A total of 94.8% of isavuconazole trough TDM samples were >1 mg/L, and 72% of levels >2 mg/L. 

## 4. Discussion

In this retrospective observational single-centre study, we show use of isavuconazole across of a range of chronic pulmonary fungal diseases, with CPA and ABPA being the most common conditions treated. Similar to previous studies, we show a significant degree of azole resistance in chronic respiratory patients with fungal disease [41]. Given the prolonged duration of antifungal therapy required, and the high prior use of azoles in patients receiving isavuconazole as a salvage therapy, based on a CLSI ECOFF of 1 mg/L, over 80.8% of *Aspergillus* spp. isolates were considered non-wild-type (resistant to isavuconazole). Given the good correlation of isavuconazole MIC with voriconazole, our study suggests that voriconazole MIC can confidently be used to screen for isavuconazole sensitivity with resultant beneficial cost implications [42]. Further studies are needed to evaluate application to non-*Aspergillus fumigatus* species, such as *Rasamsonia*, *Scedosporium,* and *Exophiala* spp.

Within our centre, previous azole intolerance was the most common indication for isavuconazole use. Post-marketing surveillance and previous studies have shown high rates of intolerance to first-line azole therapy, such as voriconazole and itraconazole [29,36,43]. Despite this, a reasonably high tolerance of isavuconazole was seen in our study (~66.6%), suggesting a role even when previous azole intolerance has been encountered. Increasing age was the strongest predictor for isavuconazole AE, with male sex also showing increased AE. Larger studies including pharmacogenomics will be needed to further explore and understand risk factors for azole intolerance, which is a significant handicap to prolonged use in chronic fungal disease. Given our results, however, cautious dosing may be considered in elderly patients starting isavuconazole, especially if there is a background of previous triazole intolerance, given the good absorption observed during the course of this study. 

Of note, significant hair loss requiring treatment interruption was not observed in any of the patients in our study, despite its having been a feature in a number of patients previously on triazole therapy. Although the exact mechanism of azole-mediated hair loss is unclear, voriconazole and itraconazole have been found to inhibit CYP26A1-mediated hydroxylation of retinoic acid in vitro at concentrations of >1 μM [44], with alopecia reported due to itraconazole, voriconazole, and posaconazole. Resultant hair loss due to azoles can cause many patients to stop treatment due to the negative impact on self-esteem, mental health, and social interactions. Our results cautiously suggest that, in circumstances where patients experience hair loss with itraconazole, voriconazole or posaconazole, isavuconazole may be a good alternative given its reduced CYP inhibition.

A potential advantage of isavuconazole compared to other azoles is its improved pharmacokinetic properties. In our retrospective cohort study, a significant proportion (86%) of individuals treated with isavuconazole were able to achieve levels >2 mg/L within the study period. This is in contrast to previous studies [10] with high prevalence of subtherapeutic levels where itraconazole, voriconazole, and posaconazole were used [7]. Out of the 275 isavuconazole TDM levels taken from this cohort, 67% were within a target trough level of between 2–5 mg/L [45]. Supratherapeutic levels were rarely noted and were not associated with an increased propensity towards adverse events. Of note, however, was a need for dose change from manufacturer’s dose recommendations to achieve a drug level above an ECAST ECOFF value >2 mg/L in 9.3% of individuals, highlighting the importance of isavuconazole TDM in a chronic respiratory fungal disease setting with high prevalence of antifungal resistance. Increased dosage was well tolerated and not associated with increased adverse events. 

At present, isavuconazole is primarily used as a salvage therapy. We have, however, recently shown the high rate of subtherapeutic azole dosage in a real-world chronic respiratory fungal disease setting, alongside the lack of integrated antifungal stewardship [46]. In this study, we highlight the improved bioavailability and tolerability of isavuconazole with reduced drug interaction. Similar or higher concentrations of isavuconazole have been observed in lung tissue compared to the blood of animal models with disseminated fungal disease. In these models, isavuconazole has been shown to reduce the fungal burden in lungs, suggesting that adequate treatment levels were achieved at this site [47]. Further multicentre, prospective, longitudinal studies are required to understand whether earlier treatment with better-tolerated azole antifungal therapy with improved bioavailability, especially in patients with unpredictable bioavailability, would lead to reduced or delayed antifungal resistance acquisition/emergence and improved outcomes.

As a retrospective observational single-centre study, our study has several limitations. Given the absence of isavuconazole from commercially available antifungal susceptibility testing kits, our findings are based on a small number of isavuconazole MIC values, and the results need to be replicated in larger multicentre observational chronic pulmonary fungal disease cohort studies. We also need better PK/PD profiling of isavuconazole as in our study there were insufficient data points to perform Monte Carlo simulations and determine a target trough/MIC index, or to evaluate a relationship with this index and efficacy. Given the high rates of antimicrobial resistance seen with salvage isavuconazole therapy, the use of a trough/MIC index may be useful to optimise azole dosing. Triazole area under the curve (AUC)/MIC ratio is the standard pharmacodynamic index associated with treatment effect [48,49]. However, AUC is impractical to measure in a clinical setting as it requires measurement of drug concentration at a minimum of two timepoints: a C_min_ and C_max_. Trough/MIC ratios have been explored as a surrogate for AUC/MIC. Exposure/response studies have been carried out with voriconazole using a trough/MIC ratio [50], where Monte Carlo simulations determined a trough/MIC ratio of 2 to 5 to be a target for voriconazole therapeutic drug monitoring. Further analysis will be needed to determine what role trough/MIC ratios will play in clinical practice in optimising isavuconazole therapy.

In summary, our study highlights the current use of isavuconazole as a salvage therapy in chronic pulmonary fungal diseases with high azole resistance prevalence. The monitoring of therapeutic drug levels underpins dose optimisation and antifungal stewardship. Isavuconazole MIC demonstrated good correlation with voriconazole MIC, and thus, the latter can be used as a surrogate marker for isavuconazole susceptibility. Positive treatment outcome or stable disease was seen in a majority of those who tolerated isavuconazole. Isavuconazole is proving to be an important new addition to the repertoire of available antifungals, showing good bioavailability and tolerability compared to other triazoles used with chronic pulmonary fungal infections. Large multicentre randomised controlled studies are, however, required to directly assess whether early use of azole therapy with better bioavailability, reduced drug interactions, and improved pharmacokinetics can reduce antifungal-resistance development in chronic pulmonary fungal disease where prolonged therapy is often required. 

## Figures and Tables

**Figure 1 jof-08-00362-f001:**
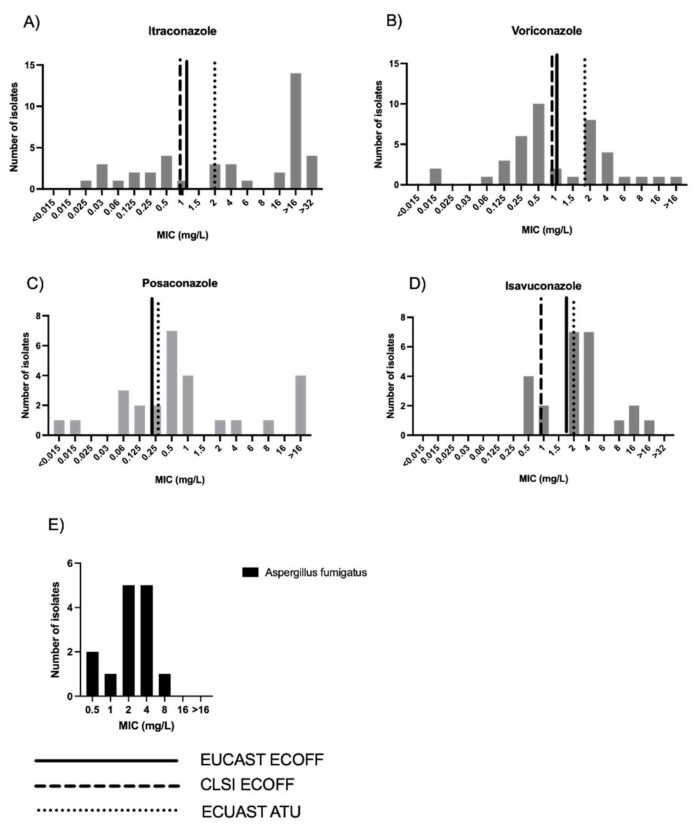
(**A**) Distribution of triazole MIC against all *Aspergillus fumigatus* clinical isolates for itraconazole during study period; (**B**) Distribution of triazole MIC against all *Aspergillus fumigatus* clinical isolates for voriconazole during study period; (**C**) Distribution of triazole MIC against *Aspergillus fumigatus* clinical isolates for posaconazole during study period; (**D**) Distribution of triazole MIC against *Aspergillus fumigatus* clinical isolates for isavuconazole during study period; (**E**) Distribution of isavuconazole MIC (mg/L) for *Aspergillus fumigatus* isolates only in individuals who received isavuconazole during the study period (total of 14 isolates). Values higher than the ECOFF values are considered to be resistant. EUCAST = European Committee for Antimicrobial Susceptibility Testing. ECOFF = epidemiologic cut-off values. ECOFFs are MICs or disk diffusion zone diameters that separate organisms into those “with and without phenotypically detectable resistance’’. ATU = area of technical uncertainty.

**Figure 2 jof-08-00362-f002:**
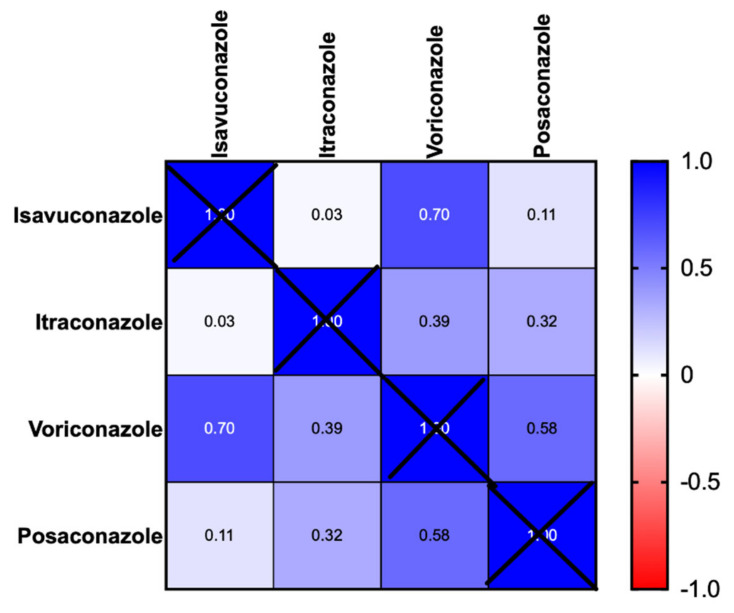
Correlation matrix of triazole MIC (Minimum Inhibitory Concentrations) for all clinical isolates throughout the study period.

**Figure 3 jof-08-00362-f003:**
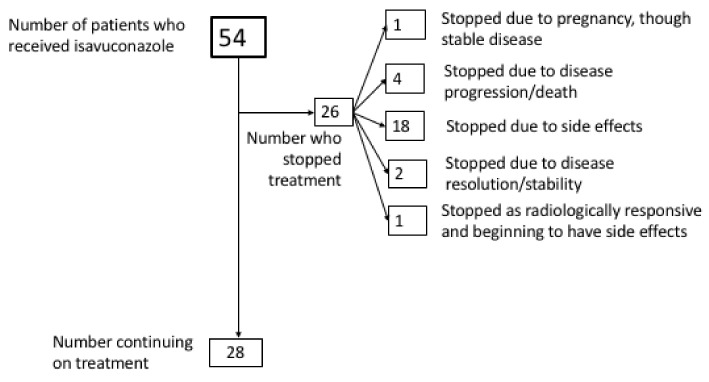
Outcomes of individuals treated with isavuconazole for chronic pulmonary fungal disease throughout the study period (*n* = 54).

**Figure 4 jof-08-00362-f004:**
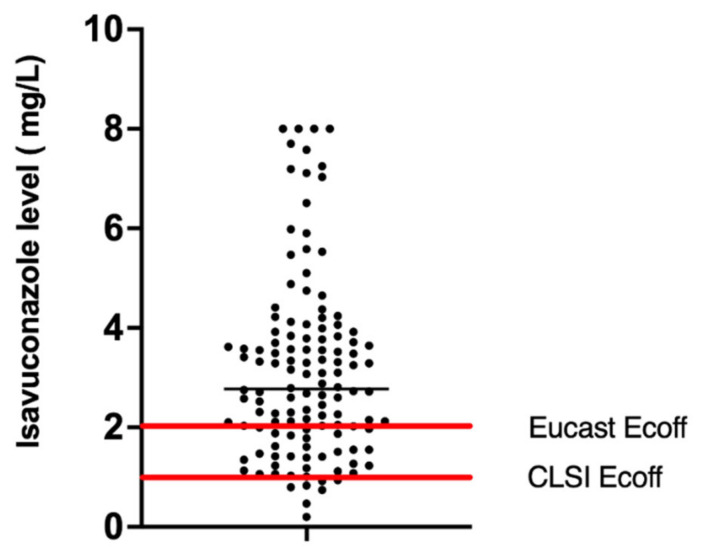
Scatter plot of distribution of isavuconazole drug concentrations. The median level is indicated by the horizontal black line, with horizontal red lines indicating the EUCAST and CLSI ECOFF for *Aspergillus fumigatus* (*n* = 132 isavuconazole drug level measurements shown).

**Table 1 jof-08-00362-t001:** Patient Demographics; *n* = 54.

Characteristic	No of Patients (%)	
Age (years)		
Mean, range	50.52, 21–82	
Sex (*n* (%))		
Male	19	(35%)
Female	35	(65%)
Primary respiratory diagnosis *n* (%))		
Asthma	11	20.0%
Bronchiolitis obliterans	1	1.8%
Cancer	2	3.6%
COPD	3	5.4%
Cystic fibrosis	19	33.9%
Immune deficiency	1	1.8%
Non-CF bronchiectasis	5	9.0%
Post-TB bronchiectasis	3	5.4%
NTM	3	5.4%
Pulmonary fibrosis	1	1.8%
Sarcoidosis	7	12.5%
Pulmonary fungal disease *n* (%))		
ABPA	18	30.0%
Chronic pulmonary aspergillosis	25	41.7%
Aspergillus bronchitis	7	11.7%
Non-Aspergillus bronchitis	6	10.0%
Pulmonary mucormycosis	1	1.7%
Aspergillus colonisation	3	5.0%
Indication for isavuconazole use (*n* (%))		
Previous triazole intolerance	34	61.8%
Prior antifungal treatment failure	11	20%
Persistently subtherapeutic alternate triazole dosing	2	3.7%
Azole resistance	2	3.7%
Drug–drug interaction	2	3.7%
Better tolerability profile	3	5.45%
Mucormycosis oral option (with intolerance of posaconazole)	1	1.81%
Distribution of isolates in patients who received isavuconazole		
Fungal species	No of isolates *n* = 53 (%)	
*Aspergillus fumigatus*	39	73.5%
*Exophiala dermatitidis*	5	9.4%
*Lomentosprora prolificans*	1	1.9%
*Rasamsonia*	2	3.8%
*Scedosporium apiospermum*	6	11.3%

ABPA: allergic bronchopulmonary aspergillosis, CF: cystic fibrosis, COPD: chronic obstructive pulmonary disease, CPA: chronic pulmonary aspergillosis, TB: tuberculosis, NTM: non-tuberculous mycobacteria.

**Table 2 jof-08-00362-t002:** Table of adverse effects experienced by patients in this study.

Patient Number	Age	Sex	Treatment Duration in Days before Discontinuation	TDM Isavuconazole (Mean or Actual) mg/L	Toxicity	Resolved to Normality on Discontinuation
1	27	M	1	No TDM available	GI toxicity	Yes
2	64	M	66	1.97	Skin reactions, hepatotoxicity	Yes
3	66	F	5	3.77	Nausea, headache, insomnia	Yes
4	20	F	284	3.67	Dizziness, fatigue	Yes
5	64	M	235	No TDM available	Hepatotoxicity	Yes
6 ^$^	60	F	72	1.94	Headache	Yes
7	49	M	7	No TDM available	Hepatotoxicity	Yes
8	66	M	6	No TDM available	GI toxicity, taste altered, appetite decreased, flu-like symptoms	Yes
9 *	54	M	489	2.48	Neurotoxicity (delirium)	Yes
10	67	F	43	3.29	Drowsiness, skin reactions	Yes
11	20	F	2	No TDM available	Hepatotoxicity	Yes
12	79	M	11	4.14	Nausea and vomiting, fatigue	Yes
13 ^$^	73	F	9	No TDM available	Fatigue	Yes
14	78	M	35	7.58	Facial swelling, difficulty in micturition, chest discomfort, rectal mucositis, ankle oedema, breathing restricted (cardiotoxicity)	No
15	65	M	16	2.17	Nausea and vomiting	Yes
16 ^$^	82	F	31	No TDM available	Appetite loss, feeling generally unwell	Yes
17 ^$^	46	F	22	No TDM available	Cardiotoxicity	No
18 ^$^	62	M	421	4.13	Skin reactions: Rash	Yes

All subjects initiated at isavuconazole 200 mg once a day except ^$^ patients (6, 13, 16, 17, and 18) who received an initial starting dose of 100 mg once a day due to frailty/previous azole intolerance. * Dose reduction made due to side effects from 200 mg once a day to 100 mg once a day.

**Table 3 jof-08-00362-t003:** Criteria/risk factors for developing adverse effects on isavuconazole treatment.

Criteria/Risk Factor	AEs *n* = 18	No AEs *n* = 34	Correlation Coefficient	Univariate *p* Value
** Dose (mg), *n*				
100 mg	6	8		
200 mg	14	23		
300 mg	0	4		
400 mg	0	1		
Age (years) (mean ± sd)	58.7 ± 18.9	46.1 ± 17.5	0.314 *	*0.021*
Sex, *n* (%)				
Male	10 (53%)	9 (47%)		
Female	8 (23%)	27 (77%)	0.3016 ^$^	*0.027*
Isavuconazole therapeutic drug level (mg/L)				
Median (IQR)	3.19 (2.13–3.56)	2.60 (1.55–3.85)	0.1035 *	0.503
Duration of therapy (days) Median (IQR)	17 (6–78)	458 (287–727)	−0.51 *	*<0.0001*

** Dose after initial TDS dose post-loading. AE: adverse effects, IQR: interquartile Range. * point biserial correlation coefficient (*r_pb_*). ^$^ Cramér’s phi coefficient = φ_c._

## Data Availability

Not applicable.

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
