# Peer review of "Experience of Isavuconazole as a Salvage Therapy in Chronic Pulmonary Fungal Disease"

_jof, 2022, doi:10.3390/jof8040362_

Round 1
Reviewer 1 Report
This is a retrospective observational study on the use of isavuconazole as a salvage agent in chronic pulmonary fungal diseases, highlighting it's efficacy and safety as best may be done in an observational study. The patient population is a difficult cohort with several comorbidities and infection with triazole resistant/ non-WT Aspergillus and other moulds. It is important data, showing clinical response in relation to TDM and MICs in such cases, though in a small sample. Otherwise well-written, it does not require any language revisions.
I have a few comments to improve clarity of analysis and result presentation:
- Statistical methods need to be elaborated to explain which tests were used to perform correlation between continuous and categorical variables, and categorical and categorical variables. e.g. Which test did you use to calculate correlation between age and adverse effects? Was age used as a continuous variable? Pearson and Spearman correlation may not appropriate for categorical-continuous correlation.
- In table 2, consider classifying adverse events into serious (those resulting in organ damage and discontinuation of therapy) and mild (managed without discontinuation of therapy). Please give unit in the column for TDM.
- Lines 238-239: What are the CIs for? If for r, then please place CIs next to r value and p value after that. i.e., [r=0.29 (95%CI 0.02,0.52), p=0.03].
- Lines 240: What do you mean by "associated with increased prevalence of adverse effects" do you mean associated with adverse effects? Using prevalence in this specific cohort may not be technically correct.
- In Table 3, please state CI of which parameter? If you mean r, then r value should also be shown.
- Line 318: Bioavailability in the lung may be claimed with serum levels. However, the authors need a reference stating serum levels are equivalent to those achieved in lung tissue for isavuconazole. Possibly https://journals.asm.org/doi/10.1128/AAC.01292-17

Reviewer 2 Report
Please read the Author's guide carefully.
Abstract: The abstract should be a total of about 200 words maximum.
・One is that there are too many characters in the abstract.
・Isn't the description method of the reference wrong?
・I am confused about the interpretation of the difference between the numbers in the text and the figure.
・I don't understand the intention of showing the median and mean together in the figure. For example, what kind of statistical processing was applied to the annotations of figures and tables, so please write the average value because it is parametric.
・The introduction is too long. Focus and shorten.
About TDM
・ How did you measure it? Do you generally measure the isavuconazole concentration in your country?
・ It seems that not all patients have undergone TDM, but what kind of patient did you choose?
・ Are there any duplicate cases of TDM? In Fig. 4, it seems that there are 132 points, so it seems that there are duplicate examples, but please indicate the blood sampling timing and intra-individual variation.
・Change the Treatment duration In months in Table 2 and 3 to Days. It is necessary to explain for each patient whether side effects occurred and it was discontinued, and how it was continued. In other words, the important information is whether the symptoms of side effects have improved after discontinuation, or whether the dose has been reduced and continued.
Round 2
Reviewer 2 Report
- If it is an in house HPLC measurement, the measurement conditions should be described. If there is a previous report, it should be cited and described.
- Table2, Treatment duration in days before iscontinuation →discontinuation
- It appears to be a trough monitor. It has a half-life of about 130 hours. Is it time for a steady state? There is no mention of the timing of the blood draw, so I do not know if it is appropriate.
